# Frequency-Enhanced Channel-Spatial Attention Module for Grain Pests Classification

**Junwei Yu** [1,*] **, Yi Shen** [2] **, Nan Liu** [3] **and Quan Pan** [2,4]

1 School of Artificial Intelligence and Big Data, Henan University of Technology, Zhengzhou 450001, China
2 College of Information Science and Engineering, Henan University of Technology, Zhengzhou 450001, China
3 Basis Department, PLA Information Engineering University, Zhengzhou 450001, China
4 School of Automation, Northwestern Polytechnical University, Xi'an 710129, China
* Correspondence: yujunwei@126.com

**Abstract:** For grain storage and protection, grain pest species recognition and population density estimation are of great significance. With the rapid development of deep learning technology, many studies have shown that convolutional neural networks (CNN)-based methods perform extremely well in image classification. However, such studies on grain pest classification are still limited in the following two aspects. Firstly, there is no high-quality dataset of primary insect pests specified by standard ISO 6322-3 and the Chinese Technical Criterion for Grain and Oil-seeds Storage (GB/T 29890). The images of realistic storage scenes bring great challenges to the identification of grain pests as the images have attributes of small objects, varying pest shapes and cluttered backgrounds. Secondly, existing studies mostly use channel or spatial attention mechanisms, and as a consequence, useful information in other domains has not been fully utilized. To address such limitations, we collect a dataset named GP10, which consists of 1082 primary insect pest images in 10 species. Moreover, we involve discrete wavelet transform (DWT) in a convolutional neural network to construct a novel triple-attention network (FcsNet) combined with frequency, channel and spatial attention modules. Next, we compare the network performance and parameters against several state-of-the-art networks based on different attention mechanisms. We evaluate the proposed network on our dataset GP10 and open dataset D0, achieving classification accuracy of 73.79% and 98.16%. The proposed network obtains more than 3% accuracy gains on the challenging dataset GP10 with parameters and computation operations slightly increased. Visualization with gradient-weighted class activation mapping (Grad-CAM) demonstrates that FcsNet has comparative advantages in image classification tasks.

**Keywords:** grain pest classification; visual attention mechanism; discrete wavelet transform; deep learning; computer vision

## 1. Introduction

Grains including cereals and legumes provide food for humans and livestock. Insect infestation is one of the leading factors affecting the quantity, quality, nutrition and market value of stored grains. Insect infestation during storage accounts for about 6–10% of postharvest grain losses, which poses serious challenges to food security in many countries [1]. In the European standards of Storage of Cereals and Pulses, ISO 6322-3 gives guidance on controlling attacks by 23 insect and mite pests. In the Chinese Technical Criterion for Grain and Oil-seed Storage (GB/T 29890-2013) [2], ten primary insect pests are specified to be identified. The species of ten primary insect pests are araecerus fasciculatus (AF, coffee bean weevil), bruchus pisorum (BP, pea weevil), bruchus rufimanus boheman (BRB, broadbean weevil), callosobruchus chinensis (CC, azuki bean weevil), plodia interpunctella (PI, Indian meal moth), rhizopertha dominica (RD, lesser grain borer), sitophilus oryzae (SO, rice weevil), sitophilus zeamais (SZ, maize weevil), sitotroga cerealella (SC, angoumois

grain moth) and tenebroides mauritanicus linne (TML, cadelle beetle). Furthermore, the unprocessed grain can be graded into basically clear grain (≤2 insects per kg), regular occurrence of insect grain (3–10 insects per kg), and intense occurrence of insect grain (>10 insects per kg), according to the population density of these ten primary insect pests. Therefore, grain insect identification and population destiny estimation are necessary for applying proper control actions.

The popular methods of insect detection and identification are visual inspection, probe sampling, acoustic detection, electronic nose and imaging methods [3]. Among them, the conventional methods such as visual inspection, trap methods and probe sampling are time-consuming and labor-intensive. Modern methods such as acoustic detection and electronic nose are costly and unreliable in noisy and complex environments. With the advancement of computer vision, image processing-based methods are proved to be more suitable for identification and classification of grain insects.

Traditional image processing methods utilize color, edge, corner, key point or other low-level features to recognize the grain pests [4–7]. For example, the United States Department of Agriculture (USDA) used visual reference images for insect detection and grain grading since 1997. Ridgway et al. [8] developed a non-touching method based on machine vision to detect saw-toothed grain beetles. Wen et al. [9] proposed a hierarchical model that combined both local features and global features to identify orchard insects.

Thanks to huge volumes of image data, convolutional neural networks (CNN) achieve great success in image classification, object detection, image segmentation and other visual tasks. CNN-based deep learning models such as ResNet [10] and VGGNet [11] have already surpassed human-level accuracy in image classification. Albeit the progress has been made in common object classifications, grain insect pest classification is still a challenging task in the practical application. As ten primary insect pests specified in GB/T 29890-2013 occur in three groups: grain weevils, grain borers and grain moths, among each group, the insects are difficult to distinguish. On the other hand, the attributes of different shapes, small sizes, multi-colors and cluttered backgrounds also pose challenges on grain insect classification. Motivated by the fact that humans and birds can find the insects in grains effectively, we introduce frequency, channel and spatial attention mechanisms into the image classification models.

This paper focuses on the frequency-enhanced attention mechanism, which integrates more clues to improve the accuracy of grain insect classification. The main contributions of this paper can be summarized as follows.

(1) We collect a challenging dataset of 10 species of stored-grain insects specified by the standard GB/T 29890-2013.

(2) We construct a novel triple-attention network (FcsNet) combined with frequency, channel and spatial attention modules. The frequency information of discrete wavelet transform (DWT) and discrete cosine transform (DCT) are involved in the convolutional neural network. FcsNet can be plugged into classic backbone networks as an efficient add-on module.

(3) Extensive experiments and ablation studies are carried out on the proposed dataset GP10 and open dataset D0. More insights into the frequency-enhanced attention mechanism can be found in the visualization results of the confusion matrix and Grad-CAM.

## 2. Related Works

In order to process the information received visually more efficiently, people are used to paying attention to some of the information while ignoring other visible information. Inspired by human vision, a new method for data processing is proposed, called attention mechanism. The attention mechanism is essential to add different weights to each part of the input information, so that the model could pay attention to areas which are more significantly weighted and thus improves the accuracy of model judgment.

To solve the problems caused by pests, Cheng et al. [12] established a system that can identify agricultural pests in a complex background using a convolutional neural

network (CNN) and residual network. This system has 98.67% accuracy for classifying the images of 10 species of agricultural pests, which is better than the ordinary deep neural network AlexNet [13]. Nanni et al. [14] proposed an automatic pest classification model by combining CNN and significance methods, but these methods [12,14] do not introduce an attention mechanism. Xie et al. [15] published a large field crop pest dataset (D0). The dataset contains about 4500 images of 40 species of field crop pests. However, the background of this dataset is single and the pose of pests is similar, which makes it easy to extract pest features. Ung et al. [16] followed a residual attention network (RAN), feature pyramid network (FPN) and a multi-branch multi-scale attention network (MMAL-Net) to improve the accuracy of the final pest classification based on integration technology and in accordance with the prediction results of the above three networks. However, they used only one attention mechanism. Zhou et al. [17] proposed an efficient small-scale convolutional neural network for pest identification, which is composed of a double fusion with a squeeze-and-excitation-bottleneck block (DFSEB block) and a max feature expansion block (ME block). Li et al. [18] developed a multi-scale insect detector (MSI_Detector) by constructing a feature pyramid to extract stored-grain insect image features with different spatial resolutions and semantic information. Shi et al. [19] proposed a multi-class stored-grain insect object detection network based on R-FCN (Region-based fully convolutional network) which achieves both high classification accuracy and speed.

In the development of attention in computer vision, common attention mechanisms can be divided into spatial attention and channel attention. Spatial attention can be viewed as an adaptive spatial region selection mechanism, and using it can directly predict the most relevant spatial locations [20,21] or select important spatial regions [22]. Hu et al. [23] captured long-range spatial context information by gather and excite operations, and they designed the GENet model, which not only emphasizes on important features, but also suppresses noise. Li et al. [24] viewed self-attention in terms of expectation maximization (EM) and proposed EM attention. Huang et al. [25] treat the self-attention operation as graph convolution and proposed cross-attention. Compared with the previous self-attention-based spatial attention [22], it improves the speed and generalization capability. Channel attention adaptively recalibrates the weight of each channel, and can be viewed as an object selection process, thus determining to what to pay attention. Hu et al. [26] proposed a new architecture unit based on ResNet [10], which is called a squeeze-and-excitation network (SENet) block. They compared the performance of global average pooling (GAP) and global maximum pooling (GMP) as squeeze operators, and finally adopted GAP to calculate the channel attention. Gao et al. [27] proposed the global second-order pooling (GSoP) block to address the limited ability of the SE block to capture global information. To overcome the high model complexity, Wang et al. [28] proposed an efficient channel attention (ECA) block. This block introduces one-dimensional convolution to reduce the redundancy of fully connected layers and obtain more efficient results. Moreover, Woo et al. [29] found that the combination of two kinds of attention has better performance through ablation experiments, and proposed the convolutional block attention module (CBAM). From another perspective, Qin et al. [30] regarded the channel representation problem in SENet as a compression process using frequency analysis, and proposed a new multi-spectral channel attention method (FcaNet) with the performance superior to that of SENet. Guo et al. [31] surveyed attention models in deep neural networks and encouraged various studies to improve deep learning results by using attention mechanisms.

## 3. Materials and Methods

### 3.1. Residual Networks

He et al. [32] proposed a residual network (ResNet) in 2015. This network solved the network degradation problem caused by too many hidden layers in the deep neural network (DNN) (this degradation is not caused by overfitting), abandoned the dropout and used Batch Normalization (BN) for training acceleration. In addition, it introduced the shortcut connection between the input and output to avoid gradient disappearance and

explosion in the DNN training. After these problems are solved, the depth of the network rose by several orders of magnitude.

The structure of ResNet can not only speed up the training of neural networks very quickly and improve the accuracy of the model, but it is also easy to optimize. Therefore, ResNet has become the basis for many research tasks, including classification, detection and segmentation. In other words, ResNet is suitable for backbone networks.

### 3.2. Channel Attention Module

The channel attention mechanism was proposed by Hu et al. [26] in 2017. It can reallocate the feature weight on the channel based on a new "feature recalibration" strategy, which has improved effective features and suppressed invalid feature information. Moreover, Woo et al. [29] noted that the global maximum pooling (GMP) also plays a role in channel attention, and has modified it, as shown in Figure 1. All above can be summarized as follows:

$$C = F_{cbam}(X, \theta) = \sigma(W_2 \delta(W_1 GAP(X)) + W_2 \delta(W_1 GMP(X))) \tag{1}$$

where X represents the input, GAP and GMP represent the global average pooling and global maximum pooling operations, respectively, $W_i$ represents the weight of the full connection layer, and the $\delta$ and $\sigma$ distribution represents ReLU and Sigmoid functions.

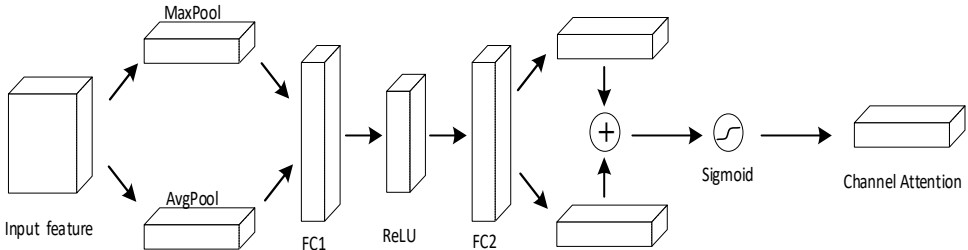

**Figure 1.** Diagram of channel attention module (CAM). As illustrated, the channel attention module utilizes both max-pooling outputs and average-pooling outputs and forward them to the fully connected layer, which finally generates channel attention through the sigmoid function.

### 3.3. Spatial Attention Module

At the same time, Woo et al. [29] noted the importance of spatial attention and proposed a convolutional block attention module (CBAM). They found that spatial attention and channel attention are complementary. Unlike channel attention, the spatial attention focuses on "where" the information part lies. In the study of spatial attention, they compared the convolution kernels of different sizes and found that a larger convolution kernel can produce better accuracy. This shows that a wider receptive field is needed in spatial attention. As shown in Figure 2, it can be written as follows:

$$S = \sigma(Conv([GAP(X); GMP(X)])) \tag{2}$$

where Conv(·) represents a convolution operation.

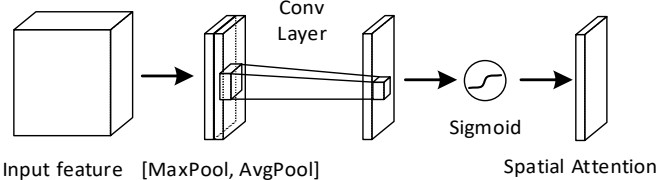

**Figure 2.** Diagram of spatial attention module (SAM). As illustrated, the spatial attention module forwards max pooling outputs and average pooling outputs to the convolution layer and generates spatial attention through the sigmoid function.

### 3.4. Frequency Attention Module

In addition to the channel and spatial attention modules, Qin et al. [30] also proposed a frequency domain channel attention network (FcaNet). Based on SENet, they regarded the channel representation problem as a compression process using frequency analysis, and analyzed GAP in the frequency domain. They mathematically proved that GAP is a special case of characteristics in the frequency domain and proposed a new multi-spectral channel attention method based on such discovery.

GAP is used to calculate the mean value of all spatial elements in each channel. However, different channels may have the same mean value, but have different semantics, which leads to poor diversity of features obtained through GAP. The discrete cosine transform (DCT) is a kind of Fourier transform and is often used to compress signals and images, and the two-dimensional DCT contains more frequency components, including the lowest frequency component GAP.

Specifically, it first divides the input images into several groups and then conducts two-dimensional DCT processing for each group. Finally, similar to SENet processing, the final weight is obtained by using the full connection layer, ReLU and Sigmoid functions. This can be written as follows:

$$S = F_{fca}(X, \theta) = \sigma(W_2 \delta(W_1[(DCT(Group(X)))])) \tag{3}$$

where DCT represents 2D discrete cosine transform while Group represents dividing the input into several groups.

Li et al. [33] found that the down-sampling (max-pooling, average-pooling and strided-convolution) in deep learning often amplifies random noise and destroys the basic results of the target. They used Discrete Wavelet Transform (DWT) to replace the down-sampling operation in the network to improve the robustness of model classification.

DWT can decompose the one-dimensional signal $s = \{s_j\}_{j \in \mathbb{Z}}$ into low-frequency components $s_1 = \{s_{1k}\}_{k \in \mathbb{Z}}$ and high-frequency components $d_1 = \{d_{1k}\}_{k \in \mathbb{Z}}$, which can be written as follows:

$$\begin{cases} s_{1k} = \sum_j l_{j-2k} s_j \\ d_{1k} = \sum_j h_{j-2k} s_j \end{cases} \tag{4}$$

where $l = \{l_k\}_{k \in \mathbb{Z}}$ and $h = \{h_k\}_{k \in \mathbb{Z}}$ are respectively low-pass and high-pass filters of the orthogonal wavelet.

If expressed by vectors and matrices, the formula (4) can be written as:

$$s_1 = Ls, d_1 = Hs \tag{5}$$

where L and H are, respectively:

$$L = \begin{pmatrix} \cdots & \cdots & \cdots & & & \\ \cdots & l_{-1} & l_0 & l_1 & \cdots & \\ & \cdots & l_{-1} & l_0 & l_1 & \cdots \\ & & & \cdots & \cdots & \end{pmatrix} \tag{6}$$

$$H = \begin{pmatrix} \cdots & \cdots & \cdots & & & \\ \cdots & h_{-1} & h_0 & h_1 & \cdots & \\ & \cdots & h_{-1} & h_0 & h_1 & \cdots \\ & & & \cdots & \cdots & \end{pmatrix} \tag{7}$$

For a 2D signal X, DWT usually performs one-dimensional DWT on each row and column, namely:

$$X_{ll} = LXL^T \tag{8}$$

$$X_{lh} = HXL^T \tag{9}$$

$$X_{hl} = LXH^T \tag{10}$$

$$X_{hh} = HXH^T \tag{11}$$

DWT decomposes an image X into high-frequency components $X_{lh}$, $X_{hl}$ and $X_{hh}$ and low-frequency component $X_{ll}$. $X_{ll}$ is the low-resolution version of the image it keeps the most energy and basic structure of the image. While $X_{lh}$, $X_{hl}$ and $X_{hh}$ represent the image details that include edges and noise. Therefore, the DWT coefficients can be integrated into the convolution neural network to extract useful features for object classification.

### 3.5. Proposed Method

In this work, we believe that channel attention, spatial attention and frequency domain attention focus on the target area in the image from different dimensions. We speculate that if these three attention modules are combined, the network's overall performance will be improved by mutual complementation. Based on the three attention modules and DWT down-sampling operation, we proposed a novel triple-attention network (FcsNet). Figure 3 shows the schematic diagram of the network we proposed.

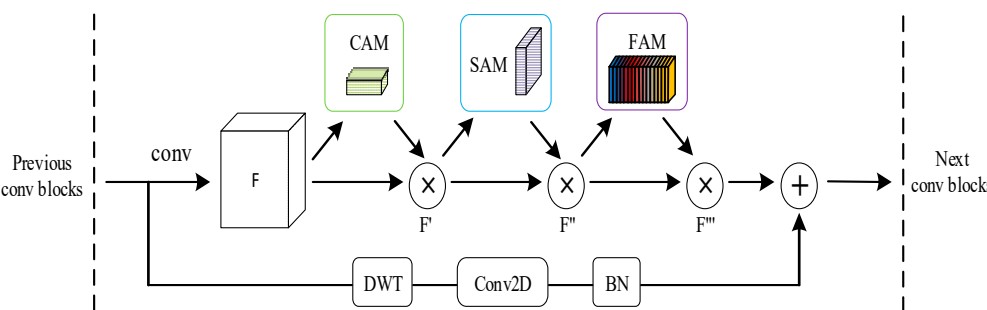

**Figure 3.** FcsNet integrated with a ResBlock in ResNet. This figure shows the exact position of our module when integrated within a ResBlock. We apply FcsNet on the convolution outputs in each block. Therein, the condition for DWT operation is Stride equal to 2.

To compare the network structures of ResNet and FcsNet (ours), we list their details in Table 1, where $DWT^1$ represents the wavelet transform substituting max-pooling operation and $DWT^2$ represents the wavelet transform substituting convolution operation with stride 2. CAM, SAM and FAM represent channel, spatial and frequency attention modules, respectively.

**Table 1.** Network structure of ResNet-50 and Fcs-ResNet-50(ours). The shapes and operations with specific parameter settings of a residual block are shown in brackets, with the numbers of blocks stacked. The right side shows different down-sampling performed by conv3_1, conv4_1, and conv5_1 with a stride of 2.

| Layer Name | Output Size | ResNet-50 | | Fcs_ResNet-50 | |
|---|---|---|---|---|---|
| conv1 | $112 \times 112$ | conv, $7 \times 7$, 64, stride 2 | | | |
| | | max pool, $3 \times 3$, stride 2 | | $DWT^1$ | |
| conv2_x | $56 \times 56$ | $\begin{bmatrix} \text{conv}, 1 \times 1.64 \\ \text{conv}, 3 \times 3.64 \\ \text{conv}, 1 \times 1.256 \end{bmatrix} \times 3$ | | $\begin{bmatrix} \text{conv}, 1 \times 1.64 \\ \text{conv}, 3 \times 3.64 \\ \text{conv}, 1 \times 1.256 \\ \text{CAM} + \text{SAM} + \text{FAM} \end{bmatrix} \times 3$ | |
| conv3_x | $28 \times 28$ | $\begin{bmatrix} \text{conv}, 1 \times 1.128 \\ \text{conv}, 3 \times 3.128 \\ \text{conv}, 1 \times 1.512 \end{bmatrix} \times 4$ | Conv2D BN | $\begin{bmatrix} \text{conv}, 1 \times 1.128 \\ \text{conv}, 3 \times 3.128 \\ \text{conv}, 1 \times 1.512 \\ \text{CAM} + \text{SAM} + \text{FAM} \end{bmatrix} \times 4$ | $DWT^2$ Conv1 $\times$ 1 BN |
| conv4_x | $14 \times 14$ | $\begin{bmatrix} \text{conv}, 1 \times 1.256 \\ \text{conv}, 3 \times 3.256 \\ \text{conv}, 1 \times 1.1024 \end{bmatrix} \times 6$ | | $\begin{bmatrix} \text{conv}, 1 \times 1.256 \\ \text{conv}, 3 \times 3.256 \\ \text{conv}, 1 \times 1.1024 \\ \text{CAM} + \text{SAM} + \text{FAM} \end{bmatrix} \times 6$ | |

**Table 1.** *Cont.*

| Layer Name | Output Size | ResNet-50 | Fcs_ResNet-50 |
|---|---|---|---|
| conv5_x | 7 × 7 | $\begin{bmatrix} \text{conv}, 1 \times 1.512 \\ \text{conv}, 3 \times 3.512 \\ \text{conv}, 1 \times 1.2048 \end{bmatrix} \times 3$ | $\begin{bmatrix} \text{conv}, 1 \times 1.512 \\ \text{conv}, 3 \times 3.512 \\ \text{conv}, 1 \times 1.2048 \\ \text{CAM} + \text{SAM} + \text{FAM} \end{bmatrix} \times 3$ |
| | 1 × 1 | global average pool, 10-d fc, softmax | |

## 4. Experiments and Results

In this section, firstly we explained our experiment. Secondly, in order to better compare our dataset (GP10) and D0 dataset [15], we rebuilt all evaluated networks [10,26,29,30] in the PyTorch framework, and used standard evaluation indicators to compare with the performance of previous methods. Finally, we studied the effectiveness of our method in the classification of grain pest images.

### 4.1. Datasets

We evaluated our proposed method on two datasets. We collected the first dataset (GP10), including 1082 pictures of 10 species of stored grain pests, namely, araecerus fasciculatus (AF, coffee bean weevil), bruchus pisorum (BP, pea weevil), bruchus rufimanus boheman (BRB, broad bean weevil), callosobruchus chinensis (CC, azuki bean weevil), plodia interpunctella (PI, Indian meal moth), rhizopertha dominica (RD, lesser grain borer), sitophilus oryzae (SO, rice weevil), sitophilus zeamais (SZ, maize weevil), sitotroga cerealella (SC, angoumois grain moth) and tenebroides mauritanicus linne (TML, cadelle beetle). Figure 4 shows some sample images of our dataset.

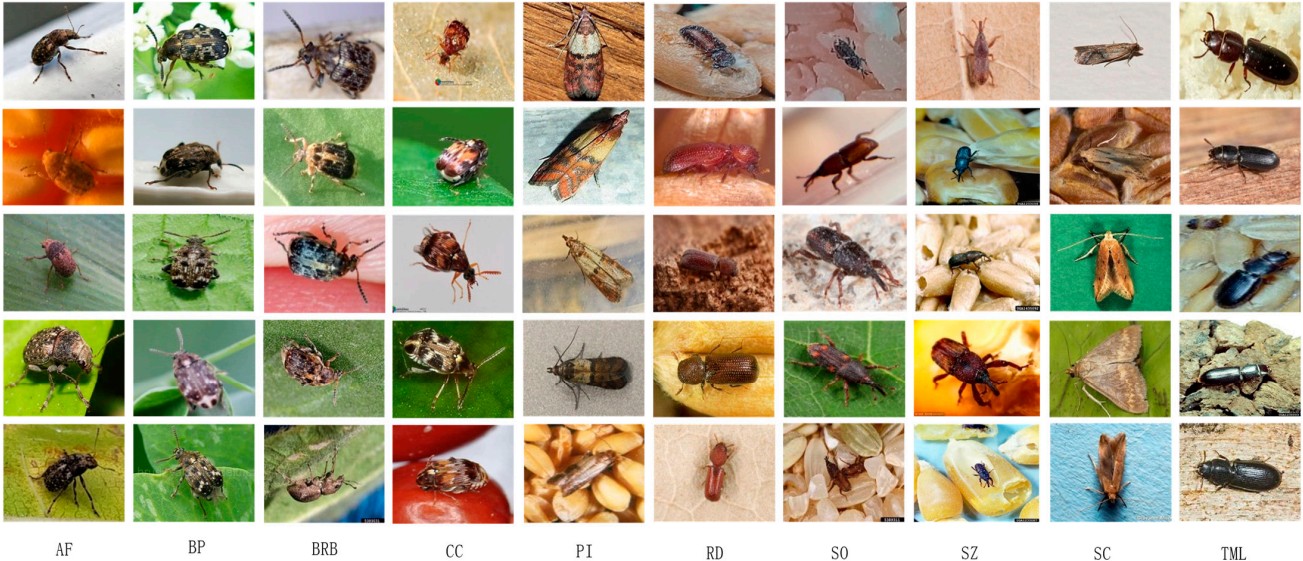

| AF | BP | BRB | CC | PI | RD | SO | SZ | SC | TML |

**Figure 4.** Sample images collected in GP10.

While collecting these samples, we relied on common image specimen search engines, including iNaturalist and Bugwood Images, etc. iNaturalist is a global community containing biodiversity data, whose goal is to promote biodiversity discipline and conservation. Bugwood Images is a funded project launched by the Center for Invasive Species and Ecosystem Health of the University of Georgia in 1994. It provides an accessible high-quality image archive and focuses on species related to economy, including insects, plants, agriculture and integrated pest management, etc.

We used the English name and corresponding synonyms of each subcategory as query keywords to search and download samples of the corresponding category. Secondly, we

searched and learned the structural characteristics of each type of stored grain pests on professional insect science websites to screen and verify each type of sample. Thirdly, we cut each type of picture according to size requirement for convenient model training later.

The second dataset is D0 (4500 pictures in all), including 40 different pests. Some are shown in Figure 5.

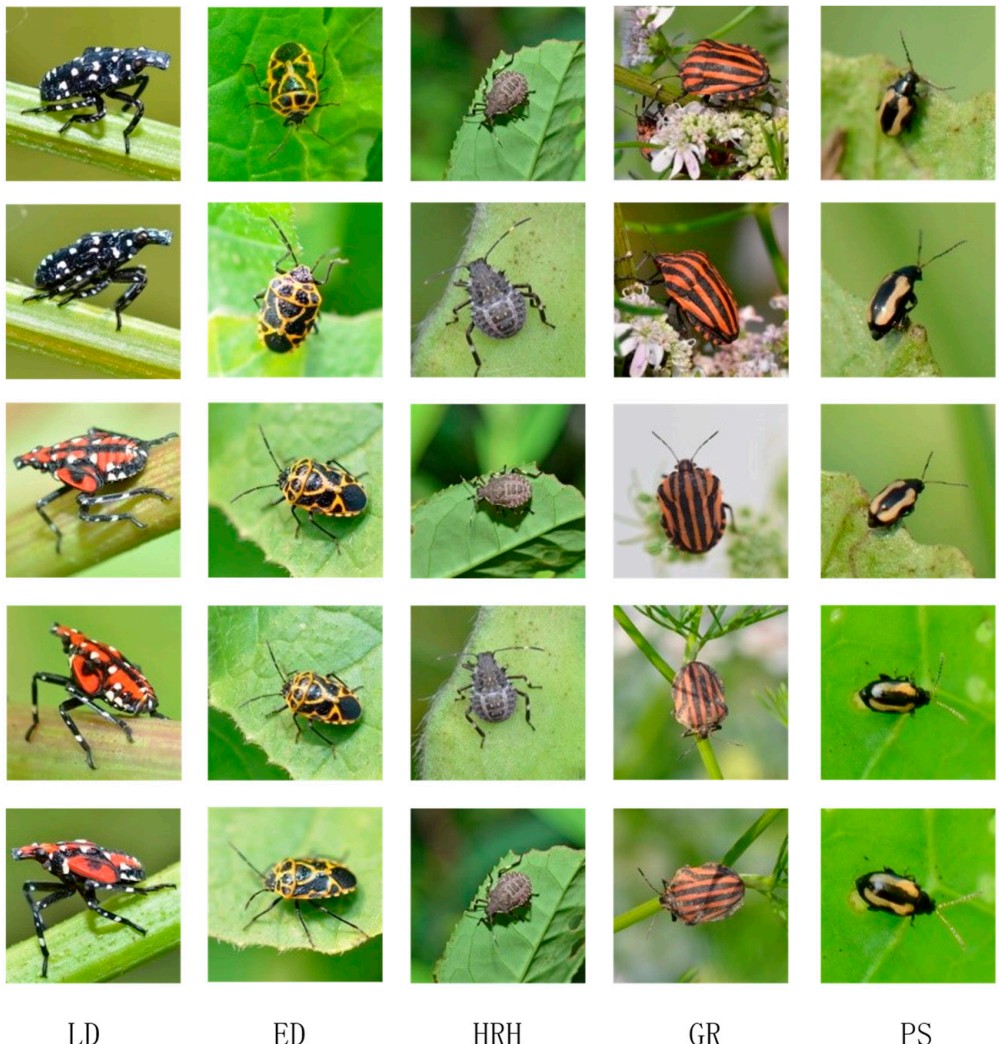

**Figure 5.** Example images in D0.

### 4.2. Experiment Settings

Our dataset is divided into three subsets: training set images (876 pcs), verification set images (103 pcs) and test set images (103 pcs), subject to the ratio of 8:1:1. See Table 2 for detailed classification. In order to obtain sufficient target features, we first expanded the training set to 2628 images by flipping horizontally and adding Gaussian noise. To make the experiment more normal and impartial, we first used python script to divide the three subsets at random, with no duplicate images present in these three subsets. The same set of division data was used in the subsequent experiments. Similarly, the same settings were used on dataset D0.

**Table 2.** Composition of the D0 dataset.

| Species | Abbreviations | Number of Samples | Train | Val | Test |
|---|---|---|---|---|---|
| Araecetus fasciculatus | AF | 115 | 93 | 11 | 11 |
| Bruchus pisorum | BP | 110 | 88 | 11 | 11 |
| Bruchus rufimanus Boheman | BRB | 97 | 79 | 9 | 9 |
| Callosobruchus chinensis | CC | 83 | 67 | 8 | 8 |
| Plodia interpunctella | PI | 129 | 105 | 12 | 12 |
| Rhizopertha dominica | RD | 69 | 57 | 6 | 6 |
| Sitophilus oryzae | SO | 176 | 142 | 17 | 17 |
| Sitophilus zeamais | SZ | 83 | 67 | 8 | 8 |
| Sitotroga cerealella | SC | 115 | 93 | 11 | 11 |
| Tenebroides mauritanicus Linne | TML | 105 | 85 | 10 | 10 |
| Total | | 1082 | 876 | 103 | 103 |

We processed the input images in advance. Firstly, we applied random clipping to the training set and adjusted its size to 224 × 224. Then, we used the method of randomly changing brightness, contrast and saturation to enhance the generalization of the model and solve the problem of overfitting. In the verification set, firstly, we adjusted the minimum edge of the image to 256, with the aspect ratio of the original image maintained. Then, we used the center clipping method to cut the image size to 224 × 224. Finally, we applied the center clipping method with the same size as the training window in the test phase. For more convenient training, we converted the data into Tensor format and standardized the data accordingly.

In the phase of training, we used the multi-class cross entropy as the cost function. Then, we used the Adam optimizer with a learning rate of $10^{-4}$ to optimize the network parameters. Next, we set the small batch to 32 and conducted 200 epochs of training. Finally, we saved the optimal training parameters and tested their predictions.

*4.3. Evaluation Metrics*

Because of the imbalanced class distribution of our dataset, we employed several comprehensive metrics for the classification task, including parameters (params), floating point operations (FLOPs), accuracy (acc), average precision (MPre), average recall (MRec), average F1-score (MF1), receiver operating characteristic (ROC) curve and area under the roc curve (AUC).

FLOPs are mainly used to describe the computation of a model, which is similar to the time complexity of an algorithm.

For the convolution kernel, we compute FLOPs as follows:

$$FLOPs\_c = 2HW\left(C_{in}K^2 + 1\right)C_{out} \tag{12}$$

where H, W and $C_{in}$ are the respective height, width and number of channels of the input feature map, K is the kernel width (assumed to be symmetric), and $C_{out}$ is the number of output channels.

For fully connected layers, we compute FLOPs as follows:

$$FLOPs\_fc = (2I - 1)O \tag{13}$$

where I is the input dimension and O is the output dimension.

Params is mainly used to describe the size of a model, which is similar to the spatial complexity of an algorithm.

The parameter number of the convolution layer is calculated as follows:

$$\text{params\_c} = C_o \times \left( k^2 \times C_i \right) \tag{14}$$

where $C_o$ is the number of output channels, $C_i$ is the number of input channels, and K is the kernel width (assumed to be symmetric). If the convolution kernel has a bias term, it will be added by one, and if not, it will not be added.

The number of parameters of the full connection layer is calculated as follows:

$$\text{params\_fc} = (I + 1) \times O = I \times O + O \tag{15}$$

where I is the length of the input vector and O is the length of the output vector.

Acc is the proportion of the true positive value to the total predicted value among all classes as follows:

$$\text{Acc} = \frac{TP}{N} \tag{16}$$

where N is the number of samples and TP is true positive. Pre is the proportion of positive values in the total number of categories. To treat the classes as being equally important, we computed the precision for each category, then took an average of them to obtain MPre as follows:

$$\text{Pre}_c = \frac{TP_c}{TP_c + FP_c} \tag{17}$$

$$\text{MPre} = \frac{\sum_{c=1}^{C} \text{Pre}_c}{C} \tag{18}$$

where C is the number of classes. $FP_c$ and $TP_c$ stand for the false positive and the true positive of the $c - th$ class, respectively. Similarly, we computed Rec and MRec as follows:

$$\text{Rec}_c = \frac{TP_c}{TP_c + FN_c} \tag{19}$$

$$\text{MRec} = \frac{\sum_{c=1}^{C} \text{Rec}_c}{C} \tag{20}$$

where $FN_c$ stands for the false negative of the $c - th$ class. The F1 combines the MPre and MRec as a trade-off as follows:

$$\text{MF1} = 2\frac{\text{MPre} \cdot \text{MRec}}{\text{MPre} + \text{MRec}} \tag{21}$$

In addition, the ROC (receiver operating characteristic) curve is used to compare the classification performance of the models. The vertical axis of the ROC curve represents the true-positive rate (TPR), and the horizontal axis represents the false-positive ratio (FPR). The higher the TPR and the lower the FPR, the better the performance of the model. In other words, the closer the ROC curve is to the upper left corner, the higher the model prediction results. TPR and FPR are defined as follows:

$$\text{TPR} = \frac{TP}{TP + FN} \tag{22}$$

$$\text{FPR} = \frac{FP}{TN + FP} \tag{23}$$

where TP, FP, FN and TN refer to true positive, false positive, false negative and true negative, respectively. The ROC curve is difficult to distinguish the performance gap between models, so we choose AUC (area under the roc curve) as the evaluation metric. The

AUC is between [0, 1], and the closer its value to 1, the better the classification performance of the model. The AUC definition is as follows:

$$AUROC = \int TPRd(FPR) \tag{24}$$

### 4.4. Experimental Results

4.4.1. Verification on Private Dataset

In accordance with the evaluation criteria in Section 4.3, we first compare the performance and efficiency of the proposed model with existing attention mechanisms on the dataset GP10 and D0, then report the results in Table 3. We observed that our method performs best on Acc, MPre, MRec and MF1. FcsNet achieves 11.65%, 9.71%, 5.83% and 3.89% accuracy gain than ResNet, SENet, CBAM and FcaNet, respectively. This means that our method is effective. This method can combine the attention of frequency domain, channel and space, and use DWT for down-sampling to improve the accuracy significantly.

**Table 3.** The performance comparison of different networks on GP10 and D0 datasets.

| Architecture | Backbone | Params | FLOPs | GP10 | | | | D0 | | | |
|---|---|---|---|---|---|---|---|---|---|---|---|
| | | | | Acc | MPre | MRec | MF1 | Acc | MPre | MRec | MF1 |
| ResNet | ResNet-50 | 23.53 M | 4.12G | 62.14 | 64.74 | 61.17 | 61.71 | 96.08 | 96.50 | 95.61 | 95.82 |
| SENet | ResNet-50 | 26.04 M | 4.13G | 64.08 | 69.30 | 64.62 | 63.93 | 97.00 | 97.49 | 96.79 | 97.00 |
| CBAM | ResNet-50 | 26.05 M | 4.14G | 67.96 | 71.37 | 67.16 | 67.45 | 97.47 | 97.76 | 97.28 | 97.40 |
| FcaNet | ResNet-50 | 26.04 M | 4.13G | 69.90 | 69.88 | 68.77 | 68.06 | 97.63 | 98.19 | 97.62 | 97.81 |
| FcsNet(ours) | ResNet-50 | 28.56 M | 5.18G | 73.79 | 74.38 | 72.79 | 71.99 | 98.16 | 98.49 | 98.33 | 98.34 |

Furthermore, we analyzed the complexity of this method from two aspects such as learnable parameters (Params) and floating point operations per second (FLOPs). For parameters, our method increased by 9.6% and 9.7%, respectively, compared with CBAM and FcaNet. For the FLOPs, our method increased by 25.4% and 25.1%, respectively, compared with CBAM and FcaNet.

Our method (FcsNet) achieved a confusion matrix as shown in Figure 6. It can be found that obvious errors are caused by several similar categories which belong to the same genus and have many common features. For example, BP and BRB belong to the same genus of bruchus, SO and SZ belong to the same genus of sitophilus.

Figure 7 shows the prediction probability of SO and SZ. Because of the similar morphology of SO and SZ, there are two prediction probabilities much bigger than the other categories. This means these two categories are often misclassified. If the top-2 error rate is considered, the accuracy will be greatly improved on the proposed dataset GP10. This also confirms that the above-mentioned categories of the same genus have common featuresand pose challenges to our research.

In order to eliminate the influence of sample imbalance, we draw the ROC curve of each model to intuitively represent the prediction ability of each model. We also calculated the AUC to make it clear which model performed better. This is shown in Figure 8. By comparison, it is easy to find that, although our model is slightly inferior to FcaNet and CBAM in the beginning, the performance of our model is slightly higher than other models in general.

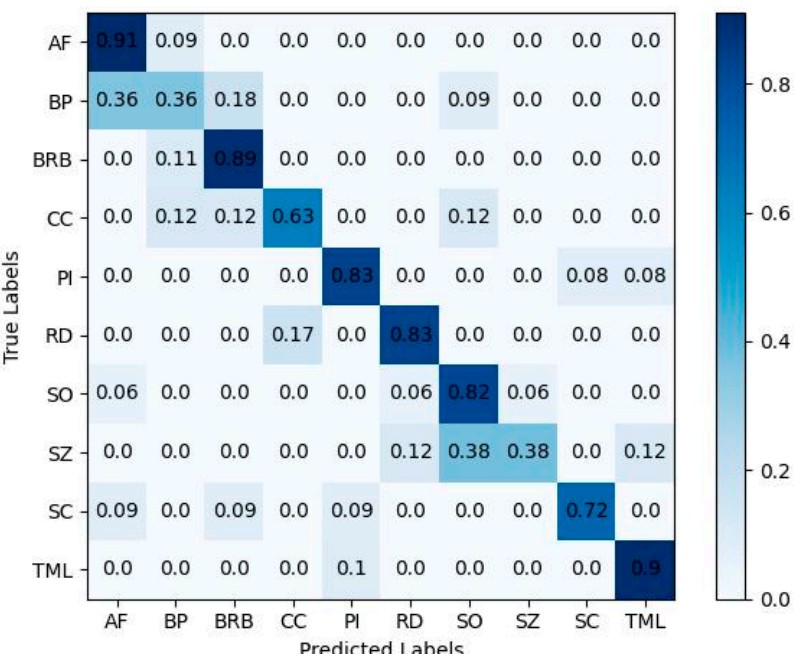

**Figure 6.** Confusion matrix of the proposed method. The vertical axis is the true label, and the horizontal axis represents the predicted label. The values in the diagonal area in the figure are the proportion of correct predictions, and the other values are the proportions of wrong predictions. The darker the color, the larger the proportion.

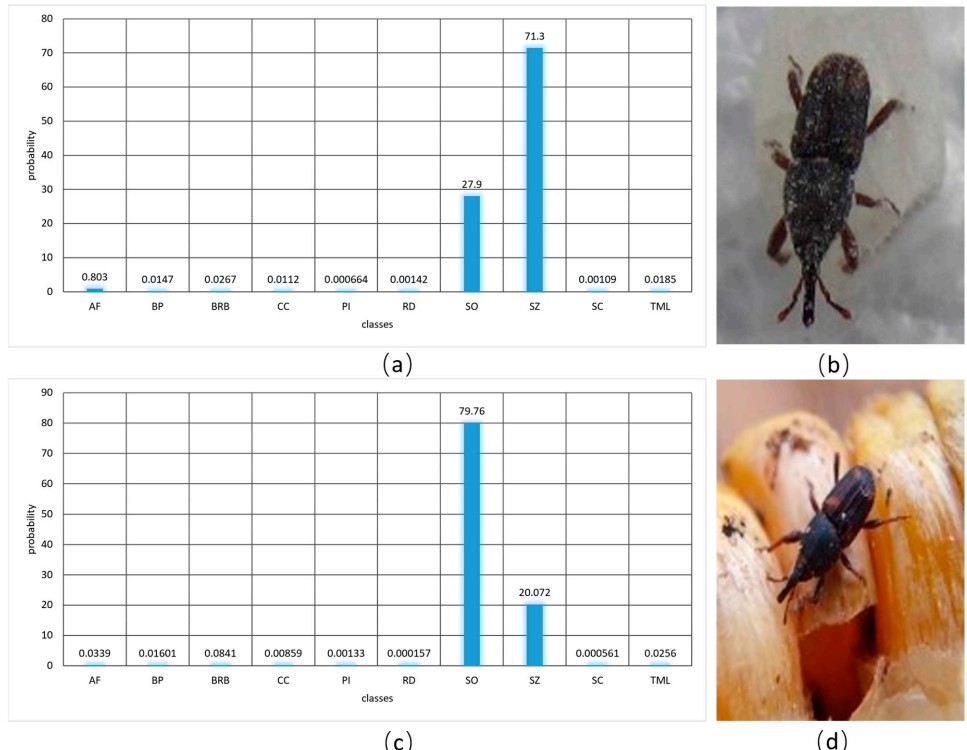

**Figure 7.** Comparison of SO and SZ prediction results, where (**a**,**c**) are bar charts of the prediction probability of SO and SZ (in percentage). The horizontal axis is the class name and the vertical axis is the probability. Examples of images for SO and SZ are shown in (**b**,**d**), respectively.

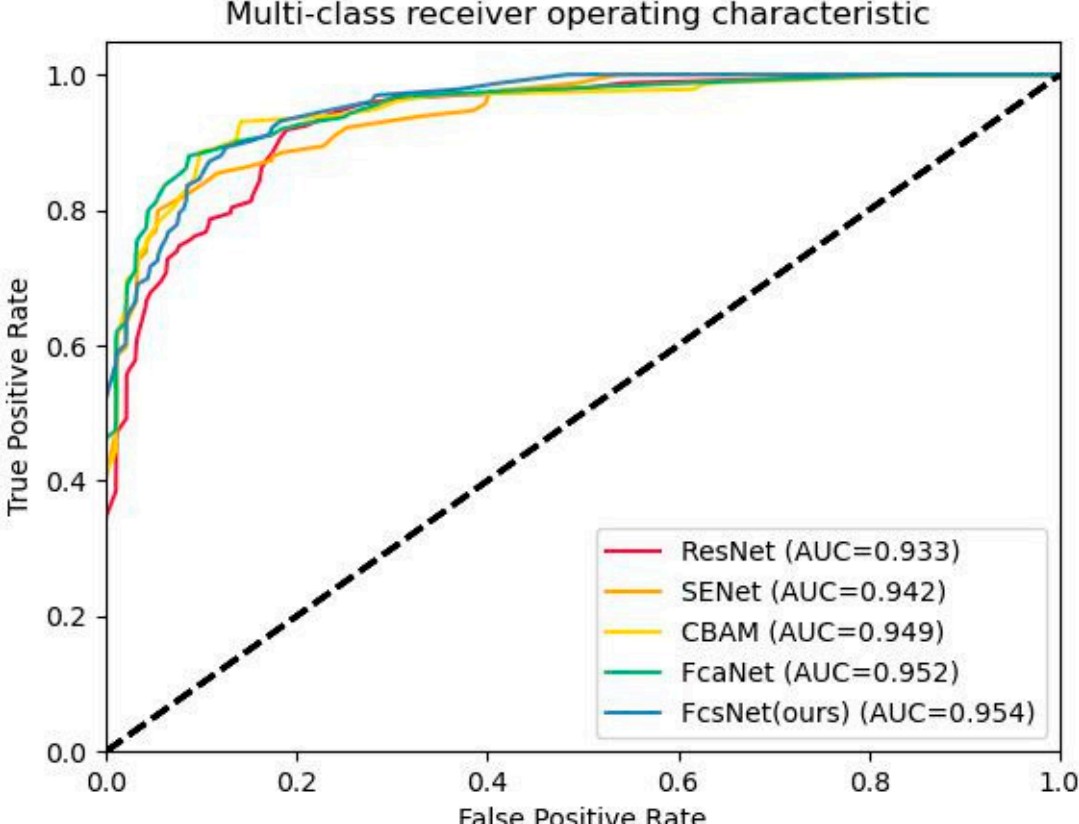

**Figure 8.** ROC comparison of different models. The horizontal axis is the false positive rate, the vertical axis is the true positive rate, and the lower right corner is the color and AUC value corresponding to the model.

4.4.2. Verification on Open Dataset

In the field of pest images, the open dataset D0 of Xie et al. [15] is often used as a standard dataset to verify proposed methods for classification. In order to further verify the performance of the proposed method, we used this dataset as supplementary proof. We observe that FcsNet is superior to other architectures in every comparison, which indicates that the benefits of FcsNet are not limited to our dataset (GP10). See Table 3 for details.

Through comparison, it is not difficult to find that the accuracy on the dataset GP10 is not as high as that on D0. Based on analysis, we concluded the following two reasons. Firstly, the images on dataset D0 have a similar background and the pest postures change slightly. In Figure 5, we give images of some categories. Secondly, our dataset (GP10) has a complex background and a high degree of similarity exists in appearance between different categories. Therefore, classification on the GP10 dataset is more challenging.

*4.5. Visualization with Grad-CAM*

This section shows the visualization of our proposed model. Previously, it was believed that the deep learning network was a black box and lacked some explanatory power, for example, in classification network models (such as VGGNet [11], ResNet [10] and MobileNet [34]), and it was unclear why the network predicted like this and where the concerns were for each category. Zhou et al. [35] proposed a kind of category activity mapping technology, which can draw a thermodynamic chart to show to which areas the network pays attention, and also where the network structure needs to be changed and retraining carried out. Moreover, Selvaraju et al. [36] upgraded and improved it based on category activity mapping to make the existing most advanced deep model interpretable without changing its architecture, thus avoiding the tradeoff between interpretability and accuracy.

Figure 9 shows the Grad-CAM [36] generated by ResNet, SENet, CBAM, FcaNet and FcsNet based on the input images of our test set. As can be seen, FcsNet includes the focus of other models in the focus input image, and it seems to focus more on the whole area of the grain pests. This also confirms the effectiveness of our proposed method.

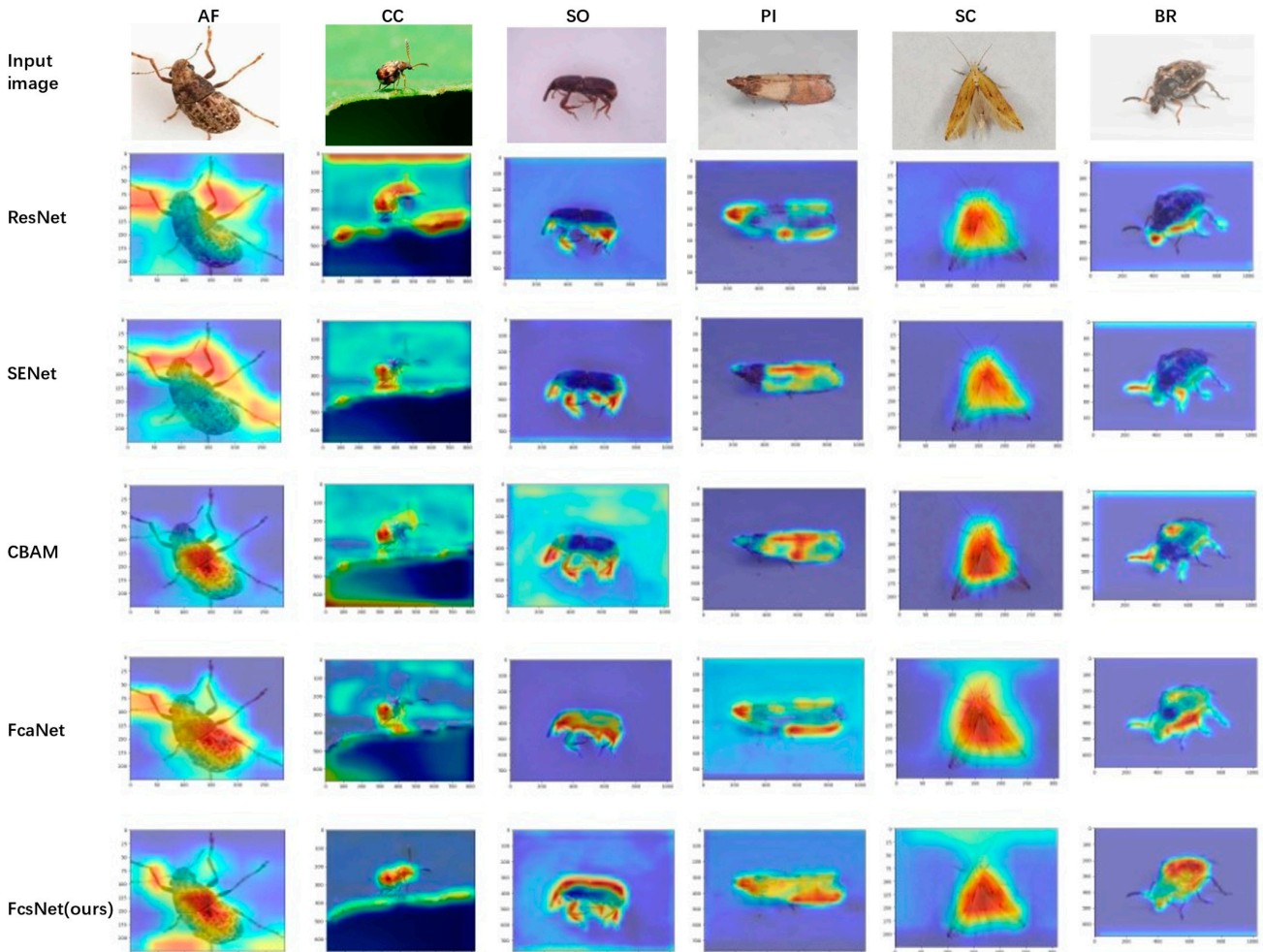

**Figure 9.** The Grad-CAM visualization results. We compared the visualization results from ResNet, SENet, CBAM, FcaNet and FcsNet, and calculated the gradient CAM visualization of the final convolution output.

## 5. Conclusions

In this paper, we propose a stored grain pest identification method based on a triple-attention module (FCS), namely, frequency domain attention (FAM), channel attention (CAM) and spatial attention (SAM). We combine the three domains and use wavelet transform for down-sampling to achieve considerable improvement in performance while maintaining a low overhead, and verified on our dataset (GP10) and D0, with the accuracy rates being 73.79% and 98.16%, respectively. FcsNet has good performance and can provide a new idea and method for the rapid detection and identification of pests. In the future, our work will focus on using multi-domain attention mechanisms to solve pest detection and segmentation tasks.

**Author Contributions:** Conceptualization, J.Y. and N.L.; Methodology, Y.S. and J.Y.; software, Y.S. and Q.P.; data curation, Y.S.; writing, Y.S. and J.Y.; writing—review and editing, Y.S. and J.Y. All authors have read and agreed to the published version of the manuscript.

**Funding:** This work was partially supported by the Key R&D and Promotion Projects of Henan Province (Science and Technology Development, 212102210152); the Innovative Funds Plan of Henan University of Technology(2021ZKCJ14); the Young Backbone Teacher Training Program of Henan University of Technology (2015006).

**Institutional Review Board Statement:** The authors are grateful to the editors and anonymous viewers for their valuable and insightful comments and suggestions.

**Data Availability Statement:** The data are not publicly available because the data need to be used in future work.

**Conflicts of Interest:** The authors declare no conflict of interest.

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
