# Peer review of "Frequency-Enhanced Channel-Spatial Attention Module for Grain Pests Classification"

_agriculture, doi:10.3390/agriculture12122046_

Round 1
Reviewer 1 Report
The is a study of proposed new stored grain pest identification method (FCS) based on multi-domain attention module, namely frequency domain attention (FAM), channel attention (CAM) and spatial attention (SA). While the study is of some scientific value, the manuscript has been poorly presented.
Literature review is not pulled together in a helpful way to show the research gap and hence the objective of the study.
The introduction/ literature has so many acronyms that need to described in first mention.
Is the dataset 1082 pictures of 10 species and output of this work? - this cannot be - if your study chose to work on this, do not limit others to the same.
Could the authors also calculate the area under curve A(AUC) for the various models studied as AUC is insensitive to imbalanced classification.
Font size and type is not the same across the manuscript
Figure 6 caption should be more elaborate than what is presented
Figure 7 can be improved - need tick marls on Y and X axis
What is reference no 7? English audience are lost here.
The writing of this manuscript need to improve interms of English.
Reviewer 2 Report
The work submitted for evaluation meets the criteria for scientific papers. As for Agriculture, it lacks a clearly defined objective and hypothesis combined with agricultural themes. The authors test a known technology on objects, of which there were 1082. I have a question why the split between the validation and test learning set is in the ratio of 8:1:1, while the norm is 2:1:1.
I would also advise you to improve the introduction to make it more accessible to the reader and develop the summary and conclusion.
The publication is generally of a good level, however, it does not bring special news to the world of science.
I recommend improving the editorial side of the work.
Round 2
Reviewer 1 Report
The manuscript has improved greatly in grammar -good job!
All querries have been satisfactorily addressed.